# Rice Fields as Important Habitats for Three Anuran Species—Significance and Implications for Conservation

**DOI:** 10.3390/ani14010106

**Published:** 2023-12-27

**Authors:** Simeon Lukanov, Andrey Kolev, Blagovesta Dimitrova, Georgi Popgeorgiev

**Affiliations:** 1Institute of Biodiversity and Ecosystem Research, Bulgarian Academy of Sciences, Sofia 1000, Bulgaria; besta.dimitrova@gmail.com; 2Faculty of Biology, Sofia University “St. Kliment Ohridski”, Sofia 1164, Bulgaria; andikolev@gmail.com; 3National Museum of Natural History, Bulgarian Academy of Sciences, Sofia 1000, Bulgaria; georgi.popgeorgiev@gmail.com

**Keywords:** agriculture, *Bufotes viridis*, *Hyla orientalis*, passive acoustic monitoring, *Pelophylax ridibundus*

## Abstract

**Simple Summary:**

Agricultural land is associated with destruction and fragmentation of natural habitats, but it can also serve as a habitat for many species. The aim of this study was to establish the abundance of three anuran species—the Marsh frog, the Eastern tree frog and the European green toad—across one natural (shallow pond) and two agricultural (rice fields) habitats near the city of Plovdiv, Central Bulgaria. Calling activity was recorded with acoustic loggers from June to September for two consecutive years. Our results indicate that all species had higher calling activity in the rice fields compared to the natural pond, and this should be considered during the process of pesticide use and approval in order to minimize the negative effect of toxic substances on anurans.

**Abstract:**

Agriculture can have different effects on wildlife depending on land-use type and extensive/intensive practices. The aim of this study was to establish the significance of rice fields in Central Bulgaria as anuran habitats. We used Audiomoth acoustic loggers to record calling activity at three sites near the city of Plovdiv—one shallow pond and two rice fields—from June to September in 2022 and 2023. We registered the three most widespread species in the region—*Bufotes viridis*, *Hyla orientalis* and *Pelophylax ridibundus*—and created pattern-matching models for them using the free web interface Arbimon, which enabled us to perform presence/absence counts and abundance estimates. At the beginning and the end of the rice vegetation period, water samples were collected and analysed for 62 substances. Substance concentrations were compared between ponds and with LC50 data from the available literature. We registered 19 and 21 substances in 2022 and 2023, respectively, with concentrations within the accepted limits, and although some metals were near risk levels, this was not reflected in the presence counts or the abundance estimates. The results indicated that frog activity was not related to any of the registered substances, but that it was positively correlated with daily rainfall and was higher in the rice fields.

## 1. Introduction

While traditional agriculture is extensive and heterogeneous, composed of different crop types mixed with patches of natural habitats, modern agriculture is a factor in biodiversity loss through the intensification of existing agricultural practices and conversion of wildlife habitats into arable land [1]. Habitat loss and fragmentation may cause population declines by decreasing dispersal and survival rates, as well as disrupting gene flow [2]. These processes result in widespread biodiversity changes that negatively affect all terrestrial ecosystems. Even though the European Common Agricultural Policy (CAP) continues to drive agricultural intensification, the application of nature-friendly farming practices is promoted by agri-environment schemes (AESs)—subsidies included in CAP Pillar 2 (rural development policy) [3]. However, AES effectiveness is limited by regional landscape features and the fact that participation in them is voluntary [4]. At the same time, in recent decades, there has been a rapid increase in the use of various chemicals in agriculture, mostly aimed at eliminating different “pest” species, or as artificial fertilizers for crops—their effects are often felt by non-target organisms that occur in the area [5]. For amphibians and reptiles, in particular, pesticides are recognised as one of the major factors contributing to population declines worldwide [6]. Despite this, there are few studies that have focused on the distribution and abundance of amphibian and reptile species in different natural and semi-natural habitats near agricultural landscapes, such as woodland patches and hedgerows, and fewer still have looked into the impact of different management regimes on herpetofauna.

Pond-breeding amphibians in temperate and Mediterranean regions live and breed mostly in temporary water bodies, and the spatial structures of their populations follow freshwater availability in their respective regions [1]. As a consequence, they could be more vulnerable to both agricultural intensification and chemical pollution compared to other species. Furthermore, wetlands in Southern Europe tend to be smaller, more temporary and more isolated than similar habitats in Northern Europe [7]. In such environments, agricultural land associated with readily available standing water, such as rice fields, can serve as an alternative to natural wetlands because a large number of invertebrate and vertebrate species use these cultivated areas for foraging and reproduction [8]. Anuran species that live in rice fields may come into both direct (through the skin) and indirect (through feeding on aquatic invertebrates) contact with the agrochemicals used to control pests or to boost productivity, but establishing the exact link between chemical use and species abundance is complicated by factors such as the great variety of different active substances in agrochemicals, their dosages and application frequencies, as well as local climatic conditions [9]. The matter is further complicated by the fact that long-term monitoring is needed in order to establish population trends at any given site. Although, over the last decade, monitoring surveys on amphibians have grown in number and have focused on diverse aims, such as status assessment and the effectiveness of conservation measures or population size estimates, they are still generally laborious and expensive (see the review in [10]).

A modern approach to population monitoring that has rapidly expanded in the last few years is passive acoustic monitoring (PAM). This method involves the use of acoustic recorders deployed in the field at specific locations and recording to a predetermined schedule. Both wildlife and environmental sounds can be collected in this way, and the resulting recordings are analysed in order to obtain useful ecological data [11]. The utility of PAM has recently been greatly improved by the emergence of new, cheap and robust autonomous recording units (ARUs) and platforms to store and analyse large amounts of audio data [12,13,14]. Among the main advantages of PAM over traditional monitoring methods is that ARUs can be positioned simultaneously in a number of study sites, which would otherwise require periodical visits of at least one trained researcher at each location [15]. Additionally, sampling during “odd” hours (e.g., during the night or very early in the morning) is facilitated and audio recordings are permanently stored, providing insights into temporal activity [11,15]. The most significant drawback of PAM is that it creates a large volume of data, with durations of recordings being usually in the hundreds or thousands of hours. The manual verification of recordings is very time-consuming; however, there are a number of developed automatic or semi-automatic techniques that speed up the process [13]. These techniques include software systems with user-friendly interfaces that also allow for more inclusive data (i.e., data that can be shared not only with academics, but also with citizen scientists, wildlife managers, enthusiasts, etc.).

Anurans are among the most vocally active vertebrate animal species, displaying a wide variety of distinctive call types [16], and as such present particularly suitable subjects for acoustic monitoring surveys. In this study, we used presence/absence data and abundance estimates derived from PAM recordings, as well as periodical water quality sampling, in order to compare the relative importance of rice fields and a natural pond in Central Bulgaria as habitats for three widespread anuran species.

## 2. Materials and Methods

### 2.1. Study Sites

The area around the city of Plovdiv in Central Bulgaria is an important agricultural region where most of the country’s rice production is concentrated. It falls within the continental climatic zone and the weather is temperate, with no dry season but hot summers (type CFa, following the Köppen–Gieger classification) [17]. We chose two rice fields of a similar size approximately 20 km from each other on a straight line, situated to the north (Rice Field 1 (R1), near the village of Tsarimir, approximate area: 3 ha) and west of the city (Rice Field 2 (R2), near the village of Tsalapitsa, approximate area: 3.5 ha). As a control site (C), we chose a stagnant pond with a natural water cycle (approximate area: 3 ha) near a large lake designated as a protected area for bird species to the north of Tsarimir and in the same area as R1. Standard rice cultivation practices in Bulgaria involve flooding of the fields in May and, after germination, which occurs in 2–3 weeks, draining and pesticide application; in late May or early June, the fields are flooded again and remain so until mid-September, when the crops are collected. For this reason, the duration of the study was set for the period June–September in both 2022 and 2023.

### 2.2. Water Sample Collection and Analyses

For both years, in the first half of June, water samples were collected from all three sites; in the first half of September, this was repeated for R1 and R2, but not for the control pond, which had become dry by that point. Each sample had a total volume of 10 litres and, for the purpose of testing for different substances, was divided into several sterile containers, following the recommendations of the certified laboratory which performed the analyses: two opaque glass containers of 2 litres each, two opaque plastic containers of 2 litres each, and four opaque glass containers of 0.5 litres each. After collection, all samples were stored in a portable fridge at 4 °C and transported to the laboratory within a few hours. The samples were analysed for the presence of 62 water quality parameters (substances), as set in the Bulgarian national standard for surface water (full lists and results of testing, as well as brief descriptions of the testing methods, are presented in a Appendix A). Comparison between means was used to determine differences between the study sites in terms of registered substances from the water samples. In addition, we compared the registered concentrations from all three sites to the available literature on the LC50 (the concentration that kills 50% of the tested animals) values of the respective substances for different species of amphibians or fish (when data for amphibians were lacking).

### 2.3. Calling Activity

From the beginning of June until mid-September, AudioMoth acoustic loggers were positioned and regularly checked at all three sites (one logger per site); using the Open Acoustic Devices AudioMoth Configuration App, the sample rate was set at 32 kHz, and all devices recorded intermittently for 10 min between 7 p.m. and 12 a.m. each day, resulting in 150 min per recorder per day. The sample rate and time period for the recordings were chosen based on the ecological characteristics of the target anuran species, all of which are most active in the evenings and early hours of the night [18]. The study focused on three target species from three families—the European green toad, *Bufotes viridis* (Laurenti, 1768) (Bufonidae); the Eastern tree frog, *Hyla orientalis* Bedriaga, 1890 (Hylidae); and the Marsh frog, *Pelophylax ridibundus* (Pallas, 1771) (Ranidae) (species taxonomy following [19]). These are the most widespread representatives of their families in Bulgaria and are known to occur in a great variety of water habitats [18]. For the analyses of the acoustic data, we used the Rainforest Connection (RFCx) Arbimon website, which is a free, cloud-based platform for storage, management, visualisation and analysis of ecoacoustic data. Using the spectrograms generated by Arbimon, three of the co-authors (SL, AK and BD) manually inspected all 1 min recordings from the first week at each site, searching for target species’ mating calls and noting presence (marked as 1) or absence (marked as 0) in each recording. During this process, frog calls with high signal-to-noise ratios were identified and selected as regions of interest (ROIs) for training sets, which were then used for pattern matching (PM) in Arbimon’s random forest models (RFMs). The training set for each species contained two ROIs and approximately 2000 min of verified presence/absence. As recommended in the Arbimon Help section, the models were validated using equal numbers of recordings with presence and absence of the respective species and had the following detection accuracies: 79% for the European green toad, 82% for the Eastern tree frog and 83% for the Marsh frog. While a number of other studies [20,21] have successfully used the simpler template PM, setting a low threshold and manually verifying portions of the results (i.e., the PM returns a high number of false positives with a reduced possibility of false negatives), we found this approach to be inapplicable in our case, as the template PM consistently returned vastly inaccurate results (e.g., >50% false positives). The merits of RFMs for species detection have been demonstrated by other studies [22], and the verified accuracy of our RFMs in all three species was considerably higher than what could be achieved using a template PM, so their results were used for the aims of the study. The RFMs were used to detect the target species in recordings from all three sites across the two years, which amounted to over 80,000 min (all files used for the analyses were uploaded on the RFCx Arbimon web platform and are available to registered users). The three study sites were compared with respect to numbers of detections in order to estimate frog activity. As the data were not normally distributed (Shapiro–Wilk *p* < 0.01), a Kruskal–Wallis test was employed, with “site” as the predictor variable, to test for differences in the number of detections between the study sites.

### 2.4. Abundance Estimates

After establishing the hour of highest calling activity in each site for the respective months, we manually went through the recordings and counted the number of calling male frogs for five consecutive days on the dates immediately before the collection of the water samples in June and September for both years. The differences in intensity and tonality of calls allow for reliable counting of the minimum number of males per site [23], and the five-day counting period was chosen so that populations could be considered as closed (i.e., with no migration, births or deaths) [23,24]. With the counts, we fitted three N-mixture abundance models [25] for the target species using the R package “ubms” [24]. The first was a null model (NULL), the second included the study site as a random effect to account for site-level heterogeneity (differences in the surrounding landscape and microhabitats) (SITE), and the third included the study sites and the values of substances with significant differences in concentrations between the three study sites (SAMPLES). Theoretical expected log pointwise predictive density (ELPD) and leave-one-out (LOO) cross-validation [26] were used for cross-validation and comparison of the candidate models.

### 2.5. Environmental Parameters

Data on the daily rainfall and air temperature in the region for the period June-September (for 2022 and 2023) were collected from the nearest automatic weather stations of the Bulgarian National Institute of Meteorology and Hydrology situated near the cities of Plovdiv and Pazardzhik (available in Bulgarian at https://www.stringmeteo.com/, accessed on 20 October 2023). The Shapiro–Wilk test was used to check for normality, and the results indicated that the data were not normally distributed (*p* < 0.01), so the Spearman rank-order correlation test was used to check for correlation between the number of detections and rainfall/air temperature.

All statistical analyses were carried out in R version 4.1.3 [27], and the chosen level for statistical significance was *p* < 0.05.

## 3. Results

For the duration of the study, we collected a total of 10 water samples (three in June 2022 and 2023 and two in September 2022 and 2023, when the natural pond was dry). Audio recordings resulted in a total of 80,124 min samples that were subsequently analysed on the Arbimon platform.

The analyses of the water samples revealed that most of the target substances were absent from all three study sites in both years; out of the total number of 62, detectable concentrations were only registered for 19 in 2022 and 21 in 2023 (see Appendix A). While all concentrations were well within the accepted Bulgarian state standards for surface irrigation water (Regulation No. 18/27.05.2009), for 12 substances, there were statistically significant differences between the study sites. However, with two exceptions, in all cases, the SAMPLES model was outperformed by the SITE or even by the NULL model (Table 1).

We were able to find published LC50 data for a total of 13 of the registered substances (Table 2). In most cases, the LC50 values were many times higher than the concentrations from our study sites (up to nearly 80,000 for antimony); however, the lowest LC50 values for three metals—magnesium, copper and manganese—were comparable to the values that were registered in the field.

Species presence was weakly related to rainfall and air temperature, with the highest values registered for the control site and the lowest for R2 (Table 3).

The Kruskal–Wallis test revealed statistically significant differences between the three sites in terms of species presence, with the percentage of detections increasing from the control site (which had the lowest values) through R1 (the percentage of detections similar to the control site but still higher) and to R2 (the highest percentage of presence for all three species) (Table 4 and Table 5). This trend was also evident from the best-fit models from the N-mixture abundance analyses (Table 5). In both years, the control site had become dry by August, so comparisons for the months of August and September were only carried out for R1 and R2. Generally, the daily calling activity for all three species over the recording period followed a similar pattern across 2022 and 2023 in the control site and R1; however, it was markedly different for the Green toad from the control site and both the Eastern tree frog and the Marsh frog from R2 (Figure 1).

## 4. Discussion

In Bulgaria, the network responsible for monitoring and quality control of the surface waters on a national level is the National System for Environmental Monitoring (NSEM). For drinking water, the maximum allowed concentrations for a number of water quality parameters are stated in Regulation No. 09/16.03.2001 and for irrigation water in Regulation No. 18/27.05.2009. All of the measured parameters from the three study sites fit comfortably within the regulations for irrigation water, which is reflected in the fact that the observed differences in concentrations between the sites had very weak support in our abundance models, and on the two occasions when the SAMPELS model had the best fit, the SITE model was a very close second (Table 1). On the whole, the SITE and SAMPLES models had very similar results, and their values were very close to each other, which in our view highlights the low significance of the measured water quality parameters of the water samples. It also has to be noted that most of the registered substances were present in all three sites, with only six exceptions—Clostridium, nitrates, nitrites, phosphates, chromium and natural uranium, the latter three of which were only registered once, in very low concentrations (see Appendix A). Although numbers of calling males may display high variation in response to biotic and abiotic factors, which are difficult to detect and to model [41], the approach we adopted accounts for detectability and is further strengthened by the fact that the counts were performed on consecutive days. The successful application of N-mixture modelling to individual frog calls for estimating male population size has been demonstrated for the Stripeless tree frog, *Hyla meridionalis* [22], and we think it is also applicable in our case.

Our results are somewhat surprising considering that other studies which were carried out in the area of R2 (Tsalapitsa) did find physiological evidence of chemical pollution in the blood cell morphology of Marsh frogs [8,42], as well as negative changes in the body condition index [43]. A study from 2017 [42] found changes in blood cells that were indicative of anaemia and disorders of the anuran immune system and hypothesized that they were caused by the presence of xenobiotics and surfactants in their living environment. However, no such substances were found in the water samples, which, similar to ours, were within the limits of Regulation No. 18/27.05.2009. The samples were collected at the same time as the frogs used for their experiments, during physicochemical monitoring conducted by the laboratory of the Directorate for Water Management of the East Aegean Region, Ministry of the Environment and Waters. This laboratory employs the same state standards for water testing as the ones used in the present study (see Appendix A), so we assume that the samples were collected and analysed in a similar manner. The authors suggest that their results were due to hypoxia, which could impair oxygen transport to tissues and is an adaptive reaction aimed at increasing frog survivability in anthropogenically altered habitats. While the authors acknowledge the apparent discrepancy between their results and the fact that the water analyses revealed no contamination, they attribute this to the lack of sediment testing. We were also unable to test sediments from our study sites, but what is certain is that, in both 2022 and 2023, Tsalapitsa had the highest percentage of presence for all three study species. All of this appears to suggest that the area provides attractive habitats for different species—at least that it did so during the two years of our study. A recent study has indicated that amphibians are not able to detect and avoid metal or metalloid contamination (particularly arsenic, iron, manganese and nickel) in otherwise suitable habitats [44], so if pollution does exist, its effect will be made even worse by the fact that so many species are attracted to the area. Indeed, studies on several Asian anuran species have revealed that rice fields attract many frogs, and there was little difference found between fields with no pesticide use and conventional ones (i.e., that use pesticides) [45]. While species abundance was higher in the “clean” fields, species diversity was higher in the conventional fields, and the authors stipulate that rice agroecosystems play a key role in maintaining amphibian populations, which in turn provide regulatory, provisioning and cultural ecosystem services [45]. While we did not detect iron in our water samples and the concentrations of arsenic and nickel were well below the recommended thresholds and reported LC50 values, the case of manganese attracts more attention. The registered values in the control pond are similar to the lowest LC50 values reported for the Australian ornamented pygmy frog, *Microhyla ornata* [37], while at the same time, they are fifteen times lower than those which have been reported for the Asian common toad, *Duttaphrynus melanostictus* [38] (Table 2). Both of these species are mainly terrestrial, breeding in shallow temporary ponds, so the extreme sensitivity of *M. ornata* is probably due to its very small size (adult frogs reach only around 2.5 cm) [37] more than any other factor; this can also explain the tolerance of *D. melanostictus*, which grows up to 20 cm [38]. In this regard, the higher manganese concentrations in the control pond are likely to have a greater negative effect on *H. orientalis* compared to the other two larger species. Still, *H. orientalis* abundance estimates were comparable to those of the other species in the control pond (Table 5), and manganese concentration did not have a discernible effect on frog abundance (Table 1). Apart from manganese, copper and magnesium also had reported LC50 values that were close to our measured concentrations (Table 2). In the case of magnesium, the authors explicitly state that the high sensitivity (around 10 times higher than previously reported) was due to the testing conditions and, in particular, the very low level of water hardness, as a reduction in the toxicity of numerous metals with increasing water hardness has been well documented [32]. The presence of calcium ions (i.e., increased hardness) led to a significant reduction in the toxicity of the tested substances, and the authors stipulate that calcium-deficient waters pose the greatest risk to aquatic life. As calcium itself is practically non-toxic to aquatic animals [31], its presence in all of the three study ponds might be beneficial for the local fauna. Sulphate and sodium ions in themselves are also non-toxic [29], so their concentrations in the study ponds are of little concern. In regard to copper, the lowest reported LC50 values are for *Ambystoma opacum* larvae, with the authors stating that anuran larvae are less sensitive, possibly because of the lack of external gills [35]. Other studies on anurans, focused on chronic exposure to copper, report good survivability (>60%) to metamorphosis at concentrations ranging from 0.01 to 0.05 mg/L and suggest that copper effects are influenced by individual-level variability among different populations [46,47]. Based on the comparison between the registered concentrations from our study sites and the available data on LC50 values for the respective substances, we can conclude that negative acute effects of any of the substances on the study species are extremely unlikely. This is corroborated by the presence data and the abundance estimates (Table 5), as well as the hourly activity data (Figure 1). The effects of chronic exposure, however, are much less clear—for example, although our models indicated no short-term significance for the high manganese concentration in the control pond, it might still negatively affect local populations in the long run.

Our study was limited in terms of the number of study sites, and we think that more large-scale surveys on higher numbers of species are needed for a better understanding of the complex relationship between agricultural practices and species abundance. Indeed, other studies have established highly variable and often non-linear relationships between biodiversity and agricultural practices [1,3,48]. However, what we have demonstrated here is that artificial wetlands (i.e., our two rice fields) are actually more populated than the nearest natural habitat. This was demonstrated by both the raw presence data and the mean abundance estimates from the N-mixture models with the best fit (Table 5). In general, the abundance estimates complement the presence data, and the slight discrepancies between them are most likely due to the fact that the presence data include all minutes with registered calls (including multiple calls from the same individual), while for the abundance estimates, only individual males were counted.

Amphibian preferences for humidity and lower temperatures (reflected in peak calling activity in the evening and during part of the night) is well-documented for all species as a whole [49], as well as for the three study species in the region, in particular [18]. In this regard, it is not surprising that species presence in all locations had a statistically significant but weak correlation with rainfall and air temperature. Unfortunately, we were not able to collect data on daily rainfall and temperature variations at the study sites, so we used data for the entire region, which might not necessarily reflect the specifics of local weather at each site. Even though the link between presence and rainfall at the control site was not very pronounced, it was still evident, especially compared to the much weaker correlation in the rice fields (Table 3). This could be explained by the fact that the water level in the rice fields is maintained artificially, while in the natural pond it is reliant on rainfall.

Regarding species presence itself, our results confirm the usefulness of PAM for monitoring anuran species in agricultural landscapes. While anuran acoustic surveys are reliable and time-efficient alternatives to traditional methods for population estimates, an obvious disadvantage is that only males are detected, so our abundance estimates reflect only male numbers. Calling males do not always signify a breeding population, and species detection probability should always be considered when estimating population size from the number of calling frogs [23]. Although during the field visits we could not carry out quantitative observation counts, we did register proportionally more individuals of all species in the rice fields, including tadpoles and juveniles. Due to their good accuracy, scalability and robustness to noise, RFMs are increasingly used in automated sound classification [50]. Recent software advancements have greatly contributed towards the automatization of analyses; however, there is still a need for manual examination of recordings and validation of models, as recommended in [21,51,52,53].

## 5. Conclusions

Our results indicate that all three species were consistently detected with greater frequency in the rice fields compared to the natural pond. Despite the limited number of study sites, our approach of combining water sample analyses with species abundance estimates provides environmentally relevant findings and presents a basis for more comprehensive future research. The importance of this type of agricultural land as anuran habitat is increased not only by the desiccation of natural ponds in the region during the summer, but also by the possibility of unintentional contamination, which could be undetectable but harmful to amphibian species. This highlights the significance of management practices on habitat suitability—while, in the current study sites, the water quality of the rice fields was similar, or even better, than that of the control pond, this could quickly change, as new pesticides are being approved with little regard to amphibian and reptile species. Future large-scale studies are needed to better understand the complex relationships in these environments. Immediate proximity to various toxic chemical substances that can potentially affect anuran survival, reproduction and physiology constitutes a serious threat to species conservation, and we advocate the implementation of pesticide approval procedures that are specifically designed for amphibians.

## Figures and Tables

**Figure 1 animals-14-00106-f001:**
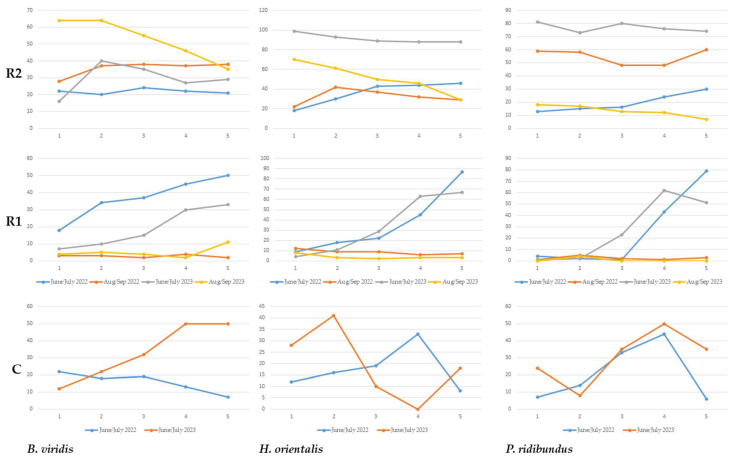
Percentages of detected presence per hour for the three anuran species in the study sites. X-axis—hour of recording, from first (7 p.m.) to fifth (12 a.m.); Y-axis—percentage of detection.

**Table 1 animals-14-00106-t001:** Predictive accuracy of candidate models of species abundance. Accuracy was measured following [26], using the expected log pointwise predictive density (ELPD) relative to the top-ranked model (ΔELPD). Data in each column are presented as the values for ELPD, ΔELPD and standard error of the ΔELPD, and the best model fit is in bold.

	Null	Site	Samples
	June 2022		
** *B. viridis* **	−143.84, −12.83, 7.171	**−131.00, 0.00, 0.00**	−133.92, −2.95, 0.86
** *H. orientalis* **	−85.74, −5.34, 3.51	**−80.40, 0.00, 0.00**	−81.19, −0.79, 0.21
** *P. ridibundus* **	**−59.87, 0.00, 0.00**	−60.55, −0.68, 0.65	−60.82, −0.95, 1.26
	**September 2022**		
** *B. viridis* **	−39.97, −16.53, 1.96	−23.81, −0.36, 0.03	**−23.45, 0.00, 0.00**
** *H. orientalis* **	−42.18, −18.73, 16.12	**−23.15, 0.00, 0.00**	−24.51, −0.42, 0.04
** *P. ridibundus* **	−52.19, −21.55, 1.94	**−30.64, 0.00, 0.00**	−31.12, −0.47, 0.49
	**June 2023**		
** *B. viridis* **	−75.90, −032, 0.78	**−75.58, 0.00, 0.00**	−76.18, −0.59, 0.43
** *H. orientalis* **	−59.53, −4.09, 3.53	**−55.44, 0.00, 0.00**	−56.24, −0.80, 0.84
** *P. ridibundus* **	−55.27, −071, 0.61	−54.68, −0.11, 0.11	**−54.57, 0.00, 0.00**
	**September 2023**		
** *B. viridis* **	−41.76, −17.42, 1.53	**−24.34, 0.00, 0.00**	−25.20, −0.86, 0.13
** *H. orientalis* **	−52.47, −19.76, 3.18	**−32.71, 0.00, 0.00**	−32.91, −0.20, 0.46
** *P. ridibundus* **	−26.05, −4.10, 0.77	**−21.95, 0.00, 0.00**	−22.42, −0.47, 0.023

**Table 2 animals-14-00106-t002:** Average concentrations of registered substances from the three study sites and their respective LC50 values (48–96 h tests) from published sources (all in mg/L). Sources that provide data specifically for amphibians are in bold and italics.

	2022			2023			LC50	Source
	C	R1	R2	C	R1	R2		
**Nitrates**	0	0	0.48	0	0	0.98	13.6–1750	** *[28,29]* **
**Nitrites**	0	0	0.016	0	0.020	0.044	33–192	** *[29]* **
**Fluoride**	0.81	0.23	0.28	0.95	0.34	0.28	51	[30]
**Calcium/Sulfates**	77	34	40	70	28	52	>2980	[31]
**Magnesium**	18	10.2	8.6	19	7	11.5	40	[32]
**Chlorides**	65	13	9.8	71	10.5	14	1178–3109	** *[33]* **
**Boron**	0.04	0.02	0.02	0.032	0.014	0.032	8.4	[34]
**Copper**	0.023	0.004	0.006	0.017	0.004	0.007	0.035–0.048	** *[35]* **
**Nickel**	0.002	0.003	0.001	0.002	0.001	0.003	0.397–0.695	[36]
**Manganese**	15.23	0.18	0.25	13.12	0.02	0.017	17–222	** *[37,38]* **
**Antimony**	0.001	0.003	0.001	0.002	0.002	0.001	238	[39]
**Arsenic**	0.019	0.003	0.002	0.014	0.003	0.002	261	** *[40]* **

**Table 3 animals-14-00106-t003:** Correlation between rainfall, air temperature and species presence in the respective sites, presented with Spearman R and *p* values (significant results are in bold and italic); *n* = 114 (number of days with rainfall and temperature data for the study period).

	Rainfall			Air Temperature		
	C	R1	R2	C	R1	R2
** *B. viridis* **	** *0.152, 0.001* **	** *0.044, 0.005* **	0.005, 0.661	** *−0.189, 0.001* **	** *−0.262, 0.001* **	** *−0.176, 0.000* **
** *H. orientalis* **	** *0.235, 0.001* **	** *0.050, 0.001* **	** *0.079, 0.001* **	** *−0.204, 0.001* **	** *−0.287, 0.001* **	** *−0.136, 0.002* **
** *P. ridibundus* **	** *0.297, 0.001* **	** *0.055, 0.001* **	** *0.057, 0.001* **	** *−0.278, 0.001* **	** *−0.257, 0.001* **	** *−0.413, 0.001* **

**Table 4 animals-14-00106-t004:** Kruskal–Wallis results on differences in species presence between the study sites; *n* = 80,123 (number of minutes analysed).

*B. viridis*	C	R1	R2
**C**		H(2) = 4.94, *p* < 0.001	H(2) = 15.04, *p* < 0.001
**R1**	H(2) = 4.94, *p* < 0.001		H(2) = 18.97, *p* < 0.001
**R2**	H(2) = 15.04, *p* < 0.001	H(2) = 18.97, *p* < 0.001	
** *H. orientalis* **			
**C**		H(2) = 3.01, *p* = 0.007	H(2) = 15.55, *p* < 0.001
**R1**	H(2) = 3.01, *p* = 0.007		H(2) = 17.39, *p* < 0.001
**R2**	H(2) = 15.55, *p* < 0.001	H(2) = 17.39, *p* < 0.001	
** *P. ridibundus* **			
**C**		H(2) = 11.36, *p* < 0.001	H(2) = 17.56, *p* < 0.001
**R1**	H(2) = 11.36, *p* < 0.001		H(2) = 28.07, *p* < 0.001
**R2**	H(2) = 17.56, *p* < 0.001	H(2) = 28.07, *p* < 0.001	

**Table 5 animals-14-00106-t005:** Percentages of species presence in recordings from the three study sites across the two years and predicted abundance estimates according to the best-fit N-mixture abundance models from Table 1. Values are presented as presence % (mean abundances ± standard deviations).

	Month	C	R1	R2
** *B. viridis* **				
**2022**	June/July	16.93% (10.30 ± 3.31)	36.57% (31.68 ± 5.90)	22.64% (39.77 ± 6.39)
	August/September	-	2.63% (3.54 ± 2.02)	45.75% (32.09 ± 7.15)
**2023**	June/July	30.73% (25.80 ± 5.80)	18.98% (37.09 ± 6.48)	29.49% (39.05 ± 6.68)
	August/September	-	5.00% (3.25 ± 1.98)	54.36% (30.27 ± 6.93)
** *H. orientalis* **				
**2022**	June/July	18.95% (20.79 ± 4.72)	36.10% (39.20 ± 6.33)	30.18% (39.19 ± 6.51)
	August/September	-	8.51% (30.57 ± 7.69)	41.14% (33.14 ± 7.61)
**2023**	June/July	21.56% (20.25 ± 4.68)	34.86% (37.55 ± 6.30)	91.46% (39.17 ± 6.35)
	August/September	-	3.79% (4.22 ± 2.05)	53.40% (36.87 ± 6.42)
** *P. ridibundus* **				
**2022**	June/July	21.66% (30.62 ± 6.41)	25.81% (26.63 ± 5.88)	16.00% (37.78 ± 6.22)
	August/September	-	2.48% (6.22 ± 2.49)	72.92% (38.66 ± 6.23)
**2023**	June/July	28.30% (24.55 ± 5.21)	27.71% (35.03 ± 6.13)	77.10% (38.94 ± 6.43)
	August/September	-	0.86% (4.63 ± 3.02)	13.94% (21.82 ± 9.65)

## Data Availability

The raw data from the results tables of the RFMs are available in a Appendix A in .csv format. All audio recordings used for the analyses are uploaded onto the RFCx Arbimon web platform. The project is public and registered Arbimon users can view its content, incl. recordings and results of the RFMs, at the following address: https://arbimon.rfcx.org/project/plovdiv (accessed on 16 September 2023).

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
