# Peer review of "Rice Fields as Important Habitats for Three Anuran Species—Significance and Implications for Conservation"

_animals, 2023, doi:10.3390/ani14010106_

Round 1

Reviewer 1 Report (Previous Reviewer 1)

Comments and Suggestions for Authors

I reviewed the modified text and the new text are according my suggestions.

I accept the manuscript in actual form.

Author Response

We are glad that we have managed to successfully integrate the referee’s suggestions in the manuscript.

Reviewer 2 Report (New Reviewer)

Comments and Suggestions for Authors

Dear Editor and Authors,

Overall I found the manuscript well written. I suggest some minor changes. Please provide a few sentences about other rice field studies on amphibians, especially involving European or Western Palearctic species. Please adress the number of ARUs used in this study. One per site? Please also provide some information on the size of the studied pond and rice fields. Size could bias the study. Please adress any potential bias arose from size differences.

Author Response

Overall, I found the manuscript well written. I suggest some minor changes. Please provide a few sentences about other rice field studies on amphibians, especially involving European or Western Palearctic species.

A: A few sentences and a new reference were added to the discussion. Apart from the studies we have already cited, we could not find others on Western Palearctic anurans in rice fields, but there are a number of studies on Asian species, and we have referenced a comprehensive book chapter.

Please address the number of ARUs used in this study. One per site?

A: Yes, the data used for the analyses if from a single ARU per site, which is now clearly stated in the Materials and methods, section 2.3 Calling activity.

Please also provide some information on the size of the studied pond and rice fields. Size could bias the study. Please address any potential bias arose from size differences.

A: Size of the study sites was specifically chosen to be similar, and approximate area for each site is now provided in the Materials and methods, section 2.1 Study sites. 

This manuscript is a resubmission of an earlier submission. The following is a list of the peer review reports and author responses from that submission.

Round 1

Reviewer 1 Report

Comments and Suggestions for Authors

The study proposes an interesting analysis comparing the abundance of three amphibian species in two rice fields and a control natural pond. However, I think the study design is poor because there is only one pond considered a control and two "replicates" to analyze. I believe that the researchers intend to go beyond a simple phenology analysis by analyzing physical-chemical parameters of the water and weather data. However, there are few meteorological variables analyzed (min T, max T, humidity, wind,...). And why only average monthly rainfall and air temperature? The authors should analyze the daily variation of meteorological variables and the number of amphibians calling.

Other problem found in the study is that the values of contaminants or water parameters studied are not compared with the published LC50 data for those species or similar species. That the values obtained are below those suggested by the administration does not mean that they are not lethal or harmful to amphibians. The guide values are intended for humans, not amphibians. It is true that the values obtained are very low, surprisingly low for a rice crop. I would like the authors to specify the analysis methodology in a little more detail. They say that they have been made in a certified laboratory, but they do not sufficiently explain the collection methodology. If the sample is small, it is easy for concentrations to be below the detection threshold of the instruments. Furthermore, although pesticides are difficult to detect in water (much better in tissues), I am surprised that no organochlorine compound or its metabolites have been detected. I would ask that you analyze the published LC50 values for the species and add them to the discussion.

The discussion is very poor because the results are not relevant. There is too much background noise in the analyzes and the correlations obtained are very weak.

I believe that the study should be redesigned again and I suggest not accepting it until profound changes are made to its structure.

Other comments in the manuscript. 

Author Response

/

Reviewer 2 Report

Comments and Suggestions for Authors

Dear Authors,

I have carefully read the manuscript entitled "Rice fields as important habitats for three anuran species – significance and implications for conservation” by Lukanov et al

The manuscript describes the reproductive activity patterns of three anurans species at three different sampling sites. The work is very well written and carried out appropriately. I only have a few minor considerations:

Line 97-99: Please modify this sentence. As it is written, it belongs to the results section and not to the introduction.

Line 133: No other species in the selected locations? If so, why were they not included? A larger sample would provide a better understanding of population dynamics

Table A1 and discussion: What about the presence of Nitrates in R2? Despite the regulation specific for the country, the presence of nitrogen derivates (nitrates, nitrites) are indicatives of contamination and it is well known to cause alterations in blood cells. Please link the results of the Table A1 with the discussion (line 288).

Author Response

/

Reviewer 3 Report

Comments and Suggestions for Authors

The manuscript entitled “Rice fields as important habitats for three anuran species – significance and implications for conservation” investigated calling activity of three species of anuran and water quality at three sites, which provide a view in anuran conservation in agriculture landscape. Although it is not the innovative topic of rice fields in anuran habitat conservation, the manuscript is well written. Besides, manuscript is messy in organization of design and can not display the shining points of the research. I provided some comments.

 Main concerns

(1) The description of the method and data analysis you used seems too long in the Abstract section. I suggest author can shortage it and add the several key results, and highlight the implications.

(2) The manuscript highlights the significance of rice fields for anuran conservation. As far as I known it is not the innovative research topic for biodiversity and conservation field. The manuscript should excavate the light spot of your research and address it.

(3) The Materials and Methods section should be classified into several subparts, such as Study area, Field work, and Data analysis. You can refer to the structure in latest paper of “Animals”. The mixed is not clear for reader and your work description.

(4) It is okey to make the correction between rainfall, air temperature and species presence. However, you should explain the what contributions of that to your research title. And the surrounding landscapes, and micro-habitat of three sites also would affect the species and abundance of anuran, please consider and add them.

(5) The research collected and analyzed the water quality of three sites. And authors try to compare the water quality between three sites of habitat for three species. It is better to create relationship among them.

(6) The method of calling activity recoder for anuran survey have some advantages and disadvantages comparing with traditional survey, please discuss that in discussion section.

Minor points

Line 93-101, This section should edit by rescheduling to write the method you use, the work you do, and the scientific question you try to answer. The results you get and the discussion you are willing to talk should delete.

Line 116-117, What standard methodology you use in water sample? What method you refer to and how did you collect the water sample.

Line 186, The front of the results should display how long of the field work and how many of the water sample and calling activity.

Line 193, The table 1 is not clear. The reader don’t know what it intend to display. The value of the table should match with the table title.

Line 333, Glad to see the conservation implication in this section. However, it should move to the end of discussion section. This section should display the conclusion you make clearly.

Comments on the Quality of English Language The manuscript needs to be edited by the native English speaker.

Author Response

/
